# Self-Lubricating Materials for Extreme Condition Applications

**DOI:** 10.3390/ma14195588

**Published:** 2021-09-26

**Authors:** Merbin John, Pradeep L. Menezes

**Affiliations:** Department of Mechanical Engineering, University of Nevada, Reno, NV 89557, USA; merbinjohn@nevada.unr.edu

**Keywords:** solid lubricants, self-lubricating materials, self-lubricating composites, friction, wear

## Abstract

Lubrication for extreme conditions, such as high temperature, cryogenic temperature, vacuum pressure, high load, high speed, and corrosive environments, is a continuing challenge among tribologists and space engineers due to the inadequate friction and wear properties of liquid lubricants. As a result, tremendous research effort has been put forward to study lubrication mechanisms for various machine elements under challenging conditions over the past two decades. Self-lubricating materials have been most widely used for adequate lubrication in extreme conditions in recent years. This review paper presents state-of-the-art of materials for lubrication in extreme condition applications in aerospace, automotive, and power generation areas. More specifically, solid lubricants dispersed in various matrices for lubrication application were analyzed in-depth under challenging conditions. This study also reports the self-lubricating materials and their lubrication mechanisms. Finally, various applications and challenges of self-lubricating materials were explored.

## 1. Introduction

The energy loss associated with moving mechanical assemblies (MMA) is a potential problem in industrial applications. Friction and wear account for almost 30% of primary energy loss, and the corresponding financial loss has been estimated to be in billions [1]. Therefore, scholars focused on novel lubricating materials to enhance the performance and reduce the frequent replacement of mechanical components due to wear failure resulting from inadequate lubrication. The concept of lubrication can be traced back to prehistoric times when people used lubricants derived from plants and animals fat to reduce friction during relative motion [2]. The stable and effective operation of a lubricant is essential to augment the efficiency and lifetime of machinery. The demand for lubrication arises when friction and wear are hard to regulate in designated applications. Unfortunately, liquid lubricants have inferior tribological properties and cannot perform their intended tasks in challenging environments [2,3,4]. In addition, equipment working under these conditions demands stable and superior tribological properties, such as excellent anti-friction properties and remarkable wear resistance for improving the reliability and service life [5]. The demand for developing novel lubricants generally arises in conditions typically encountered in aerospace [6,7,8], automotive, power generation, and machining [9,10,11] applications. The challenging conditions include high temperature (HT), cryogenic temperature, vacuum pressure, high load, high speed, and corrosive environments [12,13,14,15,16].

The inferior tribological properties of liquid lubricants can be mitigated by designing a new type of lubricant that can provide superior tribological properties. Therefore, scholars have relentlessly explored novel lubricant technologies and searched for potential lubricating materials as a global cure for inadequate friction and wear [17]. The advent of industrialization and escalating demands for efficient lubrication in diverse applications has made researchers focus more on self-lubricating materials [18,19]. These classes of materials are widely perceived to provide excellent tribological properties in extreme conditions. Moreover, they can adapt themselves based on external conditions by changing their states, and provide the required tribological properties. The challenges in the diverse fields of application have tempted researchers to develop self-lubricating materials in recent years, increasing the number of research papers related to this field. This trend has mainly been observed from the year 2011, which is represented in Figure 1.

Furthermore, developing self-lubricating materials that provide superior lubricity for severe conditions is demanding for scholars working in tribology. Dispersing solid lubricants into various matrices, such as metal matrix, polymer matrix, ceramic matrix, and intermetallic matrix, is a commonly employed method to develop potential self-lubricating materials [18,20]. The solid lubricants that are predominantly used in self-lubricating materials include soft metals [21,22,23,24], transition metal dichalcogenides (TMDs) [25,26,27,28], metal oxides [29,30,31,32], fluorides [33,34,35,36], hexagonal boron nitride (h–BN) [37,38,39,40], and polymers [41,42,43]. The solid lubricant phase associated with the self-lubricating material forms a lubricious phase due to the tribo-chemical reaction and ultimately leads to a constant lubricant supply to the interfaces. In addition to the superior lubricity, self-lubricating materials must possess properties, such as high thermal conductivity, oxidation resistance, chemical stability, and low shear strength over the entire working regime. Solid lubricants can be applied on the machine component surface by simple methods, such as painting and burnishing. Self-lubricating coatings can be prepared using magnetron sputtering [44,45], laser cladding [46,47,48], thermal spraying [49], and vapor deposition techniques [50,51,52]. In addition, powder metallurgy is a widely used technique to introduce solid lubricants into various matrices to obtain self-lubricating materials [53,54,55].

There are a significant number of reviews on various lubricants for diverse industrial applications. However, very few reviews explore the use of self-lubricating materials exclusively dedicated to extreme condition applications. This review provides a comprehensive discussion on the friction and wear behavior of self-lubricating materials for challenging environments. Section 2 addresses the dispersing of solid lubricants in various matrices, such as metal, polymer, ceramic, and intermetallic matrices. In addition, the lubrication mechanisms of self-lubricating materials under various challenging conditions were explored in detail. Finally, the applications and challenges of self-lubricating materials are elucidated in Section 3.

## 2. Tribology of Solid Lubricants

Liquid lubricants are not capable of providing superior tribological properties in severe environments. Therefore, solid lubricants were introduced to different matrices to enhance the reliability and self-adaptability of the lubricating material, and are expected to provide adequate lubrication. The primary reason for the superior lubrication properties of solid lubricants is their ability to shear through the mating surface, and their lubricious behavior can be correlated to their layered structure. For example, solid lubricants, such as TMDs, h–BN, and graphite, possess unique lamellar structures. Therefore, these solid lubricants are widely employed as a potential reinforcement phase in self-lubricating composites, and as a coating in MMA working in challenging conditions [56]. Under the action of external force, these layers align parallel to the direction of force and slide over each other, reducing the friction between surfaces during relative motion. The following section explains the tribological behavior and properties of solid lubricants dispersed in various matrices to provide superior lubricity in extreme environments. Figure 2 represents the typical solid lubricants used for self-lubricating materials in challenging environments. Table 1 indicates the applied ranges we considered in this study based on the available literature.

### 2.1. Soft Metals

Soft metals contain multiple slip planes and exhibit unique characteristics of low CoF over broad working temperatures [63]. These characteristics are due to the inherent properties possessed by soft metals, such as low surface roughness and high viscosity. For example, Ag, Sn, Au, Pb, In, Pt, etc., are considered soft metals. In soft metals, the frictional heat developed during sliding destroys lattice defects, such as dislocations and vacancies [12]. The destruction of lattice defects leads to improper work hardening, and this mechanism is responsible for providing superior lubricity in extreme conditions. Silver is the most commonly used solid lubricant among soft metals as a reinforcement in the matrix, and silver has a high diffusion coefficient. The high diffusion coefficient helps the easy formation of the lubrication film. Thus, it provides remarkable tribological properties. Scholars conducted tribological testing under different testing conditions with silver as the reinforcement in intermetallic matrix, ceramic matrix, and polymer matrix self-lubricating materials. They observed that different compounds at various temperatures give superior lubricity and low wear rate. Figure 3 shows the stable operating temperature range of various solid lubricants.

Wang et al. [24] performed tribological studies on three different samples named NA (90Ni_3_Al–10Ag), MA10 (71.57Ni–7.29Al–4.71Cr–6.32Mo–0.12Zr–0.005B–10Ag) and MA20 (63.62Ni–6.48Al–4.18Cr–5.62Mo–0.10Zr–0.005B–20Ag) from room temperature (RT) to 900 °C. The authors revealed that between RT and 400 °C, the presence of silver in the matrix provided the lubricating effect. In addition, the authors reported that an MA20 alloy showed superior self-lubricating performance over a broad temperature regime. The usage of 20 wt. % Ag in the Ni_3_Al matrix provided a CoF of 0.2 and wear rate of 1 × 10^−5^ mm^3^/Nm to 2 × 10^−5^ mm^3^/Nm between 800 °C and 900 °C. The lubrication mechanism and CoF variation over a broad temperature regime for MA and NA alloy is shown in Figure 4. The Ag_2_MoO_4_ and NiO film formed at HT prevented the direct contact of tribo pairs and provided improved tribological properties in MA alloys. On the other hand, the Ag_2_O and NiO glaze film formed on the NA alloy has inferior HT tribological properties compared to the Ag_2_MoO_4_ and NiO film formed in MA alloys. Scholars reported that Ag_2_MoO_4_ possesses a lamellar structure with weak bonding between oxygen and silver, which can shear easily at HT and provide enhanced tribological properties [64].

Liu et al. [65] studied the self-lubricating mechanism of M50 bearings used in aviation industries under high load and HT conditions. In this analysis, the researchers developed a micro-dimple structure on M50 steel dispersed with multiple lubricants. The authors considered two sets of samples, MM-S (M50–50Sn40Ag10Cu) and MM-ST (M50–50Sn40Ag10Cu–TiO_2_). The authors revealed that micro-dimple structures filled with M50 steel have a self-adaptive lubrication ability. Furthermore, the titanium dioxide nanoparticles played a significant role above 12 N. In contrast, Sn, Ag, and Cu played a pivotal role below 12 N. The addition of TiO_2_ nanoparticles imparted enhanced wear resistance and low friction properties. The self-lubricating mechanism of the M50 steel matrix dispersed with multiple solid lubricants is represented in Figure 5.

With increases in temperature, the Sn, Ag, Cu, and TiO_2_ come out of the micro-dimples, and the wear debris that forms leads to the formation of a lubrication structure due to friction. This lubrication structure is made of a lubrication layer and a compaction layer. The compaction layer formed on the subsurface supports the lubrication layer. The lubricating layer is formed on the surface rich in Sn, Ag, and Cu, shown in Figure 5a. Under heavy loads, the lubrication layer breaks down in MM-S, and the tribological properties deteriorate. When the sample is subjected to more than 12 N load, the effect of TiO_2_ particles in the micro-dimples is prominent. TiO_2_ nanoparticles are enriched in the lubrication layer, and the high strength of these nanoparticles prevents the rupture of the lubrication layer on the worn surface, as is represented in Figure 5b.

Shi et al. [22] conducted tribological testing on a TiAl matrix reinforced with 0 wt. %, 5 wt. %, 10 wt. %, and 15 wt. % silver, respectively named T1, T2, T3, and T4. The tribological testing was performed at RT and temperatures ranging from 200 °C to 800 °C. They summarized that silver provides lubrication in the moderate temperature range. Ti_2_AlC, silver oxides, titanium oxides, and silicon oxides provide a superior lubricating effect at HT, and Al_2_O_3_ acts as the wear-resistant phase. T1 and T3 samples showed higher CoF and wear rates with the rise in temperature due to the existing tribo-film thickness. The increased thickness of the tribo-film eventually leads to fragmentation and further clogging between the surfaces. Among all the tested samples, T3 exhibited better self-lubrication characteristics. The observed CoF and wear rate with the rise in temperature from RT to 800 °C for T3 samples were 0.26–0.43 and 1.56 × 10^−4^ mm^3^/Nm–3.26 × 10^−4^ mm^3^/Nm, respectively. The authors summarized that the CoF and wear rate for a TiAl matrix containing silver is less than that of the base alloy. Shen et al. [66] conducted HT tribological studies on TiAl composites containing Ag and V_2_O_5_ nanowires. The authors showed that the TiAl matrix dispersed with 5 wt. % Ag and 1.5 wt. % V_2_O_5_ nanowires have excellent tribological properties at HT due to continuity in the lubrication film and the synergic effect of Ag and V_2_O_5_. The wear mechanism of TiAl–5Ag–1.5V_2_O_5_ from 300 °C to 600 °C tested against Si_3_N_4_ is represented in Figure 6.

At 300 °C, less Ag and V_2_O_5_ is squeezed from the matrix, leading to the formation of a very minute film on the worn surface, represented in Figure 6a. When the temperature rises to 450 °C, a thick and continuous lubricating film is observed on the worn surface. In this case, the greater amount of V_2_O_5_ homogenously distributed over the film provides the shear strength and prevents the plastic flow of Ag, as shown in Figure 6b. The lowest CoF and wear rate was reported at 450 °C. When the temperature rises to 650 °C, V_2_O_5_ gets softened, and quickly forms wear debris, and it can easily detach from the worn surface. It is observed that there was no V_2_O_5_ present in the worn track. A significant amount of Ag squeezed from the wear track helps form the lubricating film at this temperature. The formed lubricating film consists of Ag and a high amount of V_2_O_5_, providing low friction and low wear, as shown in Figure 6c. Table 2 represents the various solid lubricants in intermetallic matrixes and their corresponding behaviors under various extreme conditions.

In summary, soft metals contain multiple slip planes, which prevent work hardening during relative motion. Among soft metals, silver is most widely adopted for use with self-lubricating materials for challenging environments. This is attributed to its better oxidation resistance and high thermal conductivity, which helps the easy dissipation of frictional heat during relative motion. The primary mechanism of self-lubrication for silver is the high diffusion and easy formation of the tribo layer at the interface. As a result, silver is capable of providing superior lubrication properties below 500 °C. In addition, 5 wt. % to 25 wt. % silver additions are beneficial to obtain superior tribological properties. On the other hand, a silver addition beyond 30 wt. % in the matrix leads to inferior tribological properties.

### 2.2. Transition Metal Dichalcogenides (TMD)

The general representation of these classes of compounds is MX_2_. The M can be Mo or W, and X can be S_,_ Se, or Te. The most commonly used TMDs are molybdenum disulfide (MoS_2_) and WS_2_. In MoS_2,_ the Mo atom is at the center, sandwiched between two S atom layers. MoS_2_ consists of a lamellar structure with thin atomic planes and is anisotropic [73]. The weak van der Waals forces between the interlayers help easy shearing in the <0001> crystallographic direction and in parallel basal planes [26,74,75]. The easy sliding in the <0001> direction leads to reduced CoF and improved wear resistance. In addition, the strong ionic bond exit between sulfur and molybdenum offers significant resistance against penetration of asperities. As a result, TMDs can quickly move in the direction of the applied load, thereby resulting inS minimum friction. Researchers reported that MoS_2_ is highly susceptible in a moist environment. Therefore, researchers adopted different strategies for enhancing the tribological performance of MoS_2_ in a humid environment. The adopted strategies include multilayer films and various doping elements [76,77,78]. Scholars observed that MoS_2_–Ti film and MoS_2_–Pb film possess enhanced friction and wear properties [79]. However, when the MoS_2_–Ti film was tested under a strong vacuum, it showed inferior tribological properties. Zhao et al. [27] studied the self-adaptive behavior of the MoS_2_–Pb–Ti film used for space applications. The authors summarized that the introduction of Pb and Ti improved the tribological properties of a MoS_2_–Pb–Ti film tested in vacuum and air at different relative humidity values (RH). The variations in CoF and wear rate for MoS_2_ and different films coated on MoS_2_ tested in a vacuum are shown in Figure 7.

The MoS_2_–Pb–Ti film tested in vacuum showed low friction and low wear properties because of the improvement in elastic modulus, transfer layer, and dominant contact interfaces of MoS_2,_ whereas in humid air, the sacrificial effect of Pb and Ti hindered the attack of O_2_ and H_2_O. The MoS_2_–Pb–Ti film showed a lower and more stable CoF compared to other films on MoS_2_. The reported wear rate for MoS_2_ is 8.8 × 10^−7^ mm^3^/Nm, which is the highest among all the other films. However, MoS_2_–Pb–Ti showed a reduced wear rate compared to other films. The CoF and wear rates for MoS_2_ and various films of Ti, Pb, and Pb–Ti on MoS_2_ at various RH values are represented in Figure 8. Upon deeper examination, it is clear that the CoF for MoS_2_ film increases to a higher value with heavy fluctuation when RH increases from 10% to 70% (Figure 8a). The MoS_2_–Ti film did not show much variation in CoF with an increase in RH, as shown in Figure 8b. However, the MoS_2_–Pb film showed a CoF of 0.03 at RH 10%, which jumped to 0.19 at RH 70% (Figure 8c). The MoS_2_–Pb–Ti showed fluctuations at high humidity values but similar CoF values to the MoS_2_–Ti film (Figure 8d).

The combination of MoS_2_, Pb, and Ti gives the best tribological performance compared to stand-alone MoS_2_ and a combination of MoS_2_ with Ti or Pb when tested in a vacuum and in the air in the humidity range 10–70%.

Dunckle et al. [14] conducted friction studies on MoS_2_ + Ti films under cryogenic vacuum conditions. They summarized that MoS_2_ + Ti films could retain their tribological performance with low wear when subjected to thermal cycling from RT to cryogenic temperature under ultra-high vacuum conditions. They also revealed that MoS_2_-based coatings are an effective lubrication provider in a vacuum in the cryogenic environment. Liu et al. [80] studied the friction and wear properties of M50 steel-based self-lubricating composites containing 5 wt. % MoS_2_ tested from 150 °C to 450 °C. The tribological test conditions included a ball-on-disk HT tribometer with a load of 20 N and a sliding speed of 0.2 m/s against Si_3_N_4_. They summarized the excellent lubrication performance at higher temperatures because of the enriched presence of MoS_2_ and FeS in the film, and they reported that above 250 °C, the regeneration of FeS provided significant stability to the film up to 450 °C. The wear mechanism of M50 steel with 5 wt. % MoS_2_ from 150 °C to 450 °C is represented in Figure 9. The authors revealed that a lubricating structure is absent at low temperatures, and on the worn surface, they observed small amounts of MoS_2_ and FeS, as shown in Figure 9a.

Under the external load from a Si_3_N_4_ counter material, the subsurface of the M50 steel dispersed with 5 wt. % MoS_2_ when subjected to compaction, and the compacted layer formed at 250 °C. At the same time, the MoS_2_ and FeS from this layer were squeezed out slowly to form a lubricating film. The formed film was not adequate to cover the whole friction surface, and hence at 250 °C, inferior tribological properties were observed as represented in Figure 9b. At 350 °C, enhanced lubrication properties were observed because of the presence of continuous lubrication film. The lubricating film possesses the synergistic effects of the lubrication properties of MoS_2_ and the plastic flow of FeS, as shown in Figure 9c. At 450 °C, the authors observed partial damage of the lubrication structure, the oxidation of MoS_2_ to MoO_3,_ and the reduced strength of the compaction layer, as shown in Figure 9d, which eventually led to the deterioration of the superior tribological properties. The decomposition of FeS explains the partial existence of the lubricating film. The author’s design reduces friction and wear properties for M50 steels dispersed with 5 wt. % MoS_2_ bearing steels compared to pure M50 at 450 °C. WS_2_ is another popular TMD with a layered structure that prevents the friction derived from the contact of surfaces, which it converts into a relative slip of the molecular layer under external load. Wu et al. [81] studied the dry sliding characteristics of a Ag–Cu-based composite containing 8 wt. % to 24 wt. % WS_2_ tested at three different conditions: vacuum, dry nitrogen, and humid air. The tribological test was conducted on a pin-on-disc tester with a sliding velocity of 1 m/s and a load of 5 N, and the counter body was silver, having a hardness of 120 on the Brinnel scale. The authors observed high CoF and low wear rates in the humid air, but low CoF and low wear rates for dry N_2_. Cao et al. [82] replaced graphite with WS_2_ in the copper matrix composite and observed significant improvements in the mechanical properties and wear resistance when graphite was returned, along with WS_2_, into the copper matrix. These superior properties are due to interfacial chemical bonding between WS_2_ and the copper matrix. On the other hand, the authors reported excess solid lubricant on the worn surfaces, a thicker film, and a low depth of the plastic deformation zone in the subsurface.

In summary, the most widely used TMDs for self-lubrication purposes in challenging environments are MoS_2_ and WS_2_. The lamellar structure helps to provide easy shearing during relative motion and can provide superior lubrication characteristics up to 400 °C. MoS_2_ cannot be used for self-lubrication in humid or moist environments because of the oxidation reaction. The oxide formed possesses higher shear strength and gives rise to inferior tribological properties. WS_2_ can withstand temperatures up to 800 °C and provide superior lubrication properties even at cryogenic temperatures (−190 °C).

### 2.3. Metal Oxides

Lubricious oxides (binary and ternary) are thermally stable at HT and are considered effective lubricants. However, they are not capable of providing lubrication at RT. It is mentioned that with temperature increases, a change in tribological behavior is observed for many oxides. These changes in tribological properties can be related to the brittle to ductile transition when the temperature rises above a critical temperature. Therefore, temperature plays a predominant role in oxide lubrication. Researchers have proposed several theories regarding the lubricating mechanism of oxides. The crystal–chemical model centered around the crystal chemistry of oxides has received wide attention [83]. This model primarily correlates the CoF and the ionic potential of lubricious oxides. The author postulated that the higher the ionic potential of oxides, the lower the CoF. At higher ionic potential, the cations are screened effectively by oxygen, and here, they did not come across other cations during sliding, which helps with easy shearing. However, there are controversies regarding the crystal–chemical model. Scholars have reported that oxide lubrication is complicated because some oxides can plastically deform and protect the interacting surfaces. Apart from that, many oxides break up during sliding and lead to abrasive wear. The introduction of the polarizability approach solves the issues associated with the crystal–chemical model. Under the polarizability approach, scholars have correlated the binding energy and polarizability of ions [84].

Zhu et al. [29] studied the friction and wear properties at HT of NiAl alloy and two different NiAl matrix-based composites containing metallic oxides, such as ZnO and CuO, and metallic powders of Mo and Cr. The authors observed that the NiAl alloy had significant wear and CoF at elevated temperatures. However, the NiAl composite containing solid lubricant CuO showed a reduction in CoF when tested from RT to HT. The reported value of CoF was about 1.0 at RT and 0.53 at 600 °C, with a low CoF of about 0.28 at 800 °C which rose to 0.3 at 1000 °C. The authors observed a smooth lubricating layer composed of CuO and MoO_3_ at 800 °C, responsible for the self-lubricating behavior. The wear rate increased when the test temperature increased from RT to 600 °C, and showed a reduced wear rate of 2.3 × 10^−6^ mm^3^/Nm at 800 °C.

The worn surface of the composite NiAl + 25.47 Cr + 10.07 Mo + 15.00 CuO is shown in Figure 10. The authors observed that delaminated layers at RT and grooves were visible at 600 °C, as represented in Figure 10a,b, respectively. A smooth lubricating glaze layer was observed at 800 °C, indicated in Figure 10c. At 1000 °C, the formed lubricating glaze layer was broken down, and grooves were observed, as in Figure 10d. Similar CoF and wear rates were reported for the composite containing ZnO. The superior tribological properties at 800 °C for ZnO-based composite are due to MoO_3_ and Cr_2_O_3_. At 1000 °C, the ZnO-based composite showed enhanced wear resistance due to the ZnO layer on the wear track. Essa et al. [85] studied the effect of WS_2_ and ZnO as solid lubricant additives on the friction and wear behavior of M50 steel matrix composites from RT to 800 °C. The authors considered four different composites named M (M50), MZ (M50 + 10%ZnO), MW (M50 + 10%WS_2_), and MZW (M50 + 10%ZnO + 10%WS_2_).

The tribological test was conducted on a pin-on-disc at HT with a sliding velocity of 0.2 m/s and applied load of 12 N, and the counter surface used was Si_3_N_4_. The authors revealed that the individual addition of solid lubricants does not enhance tribological properties. The authors summarized that WS_2_ helps to enhance lubricating properties from RT to 400 °C, except at 200 °C. On the contrary, ZnO imparted superior tribological properties in the temperature range of 600 to 800 °C. ZnO and WS_2_ synergistically reduced CoF and wear rate over a broad range of temperatures. The CoF was reduced by 43.64% for MZW compared to M at 800 °C. EDS analysis confirmed that ZnO and ZnWO_4_ were responsible for reducing CoF at HT for the MZW composite. Wang et al. [86] studied the friction and wear behavior of a NiAl composite coating containing nanostructured TiO_2_ and Bi_2_O_3_ from RT to 800 °C. The authors considered three composite coatings with a 3:2 ratio of TiO_2_ and Bi_2_O_3_. The authors observed a significant reduction in friction and wear properties at 800 °C. The reduced CoF and wear rate are due to the combined effect of Bi_4_Ti_3_O_12_ and NiTiO_3_. The composite coating with 30 wt. % TiO_2_ and Bi_2_O_3_ showed better low-friction and low-wear properties for the whole test temperature. Li et al. [87] studied the tribological behavior of a Ni–Cr–Mo-based composite containing TiO_2_ and Bi_2_O_3_ as solid lubricants. The authors considered four different composites (NC1, NC2, NC3, and NC4) having different wt. % TiO_2_: Bi_2_O_3_ and NiCr (NC) alloy. In the experiment, they used 0 wt. %, 10 wt. %, 20 wt. %, and 30 wt. % metal oxides (TiO_2_/Bi_2_O_3_). The authors observed lower CoF for NC3 and lower wear rate for NC4 containing 20 wt. % and 30 wt. % metallic oxides. The friction and wear studies were conducted using a ball-on-disk at HT with a load of 10 N and sliding speed of 200 r/min, against Al_2_O_3_ at 800 °C. The variations in CoF and wear rate for different samples are indicated in Figure 11. Among the five samples, NC showed the highest CoF and wear rate. However, NC1 showed a CoF of 0.36 and a reduced wear rate compared to NC. This is because of the formation of the MoO_3_ layer at HT. Increased additions of TiO_2_ and Bi_2_O_3_ directly affect the CoF and wear rate of NC2, NC3, and NC4. The increased wt. % of metal oxide in the matrix for these specimens reduced the CoF and wear rate, as shown in Figure 11a,b. NC4 showed a low CoF of 0.18. NC4 has a higher wear rate than NC3, due to the shredding of oxide particles because of friction. The remarkable tribological properties at HT are due to the formation of Bi_4_Ti_3_O_12_ on the worn surface. The authors observed a lower CoF for NC3 and a lower wear rate for NC4 containing 20 wt. % and 30 wt. % metallic oxides. Table 3 represents a self-lubricating composite based on ceramic matrix and its HT behavior.

In summary, both binary and ternary oxides can provide superior tribological properties at high temperatures. Among binary oxides, V_2_O_5_ provides a low CoF and wear rate because of the low shear strength. A combination of metallic oxides can provide superior tribological properties at extremely HT.

### 2.4. Fluorides

Fluorides exhibit significantly superior tribological properties above 500 °C, and inferior properties at low temperatures and RT [92]. The primary reason for providing potent lubricity at HT is the change in wear mechanism from brittle-to-ductile, while the higher CoF and enhanced wear rate are due to the three-body abrasion. LiF, CaF_2_ BaF_2_, CeF_3_ LaF_3,_ etc., are some of the examples of commonly used fluorides. CaF_2_ is soft, poorly soluble in water and thermally stable over a wide temperature, and its thermal expansion closely matches that of many alloys [93,94]. These properties lead to the extensive use of CaF_2_ in ferrous-based self-lubricating composites [95]. Han et al. [93] conducted friction and wear studies on Fe–Mo composites containing different wt. % of CaF_2_ content. The authors divulged that 8 wt. % of CaF_2_ in the Fe–10Mo matrix provides a superior reduction in wear and CoF at RT and at 600 °C. The reported self-lubricating characteristics for Fe–Mo–CaF_2_ at HT are due to the lubricious film consisting of CaMoO_4_ and CaF_2_. Similar results were observed when 8 wt. % BaF_2_ was added as a solid lubricant Fe–Mo-based self-lubricating composite [96]. Liu et al. [97] conducted friction and wear studies on self-lubricating cemented carbides based on WC–12Co with 0 wt. % to 20 wt. % CaF_2_ as the solid lubricant. The authors reported that the introduction of CaF_2_ caused a reduction in CoF, reduced wear loss, and refined grains of WC. A lubricating film based on CaF_2_ was formed at the interfaces during tribological testing, which remarkably increased the tribological properties. The authors reported that 20 wt. % CaF_2_ reduces the CoF by 22% and wear loss by 93%, compared to the friction and wear behavior of WC-12Co. However, they also revealed that a greater addition of CaF_2_ leads to increased porosity, which subsequently causes degradation in the mechanical properties. Figure 12 shows the volume loss and CoF at different volume percents of CaF_2_ during tribological testing. It is evident that the addition of a higher volume percent CaF_2_ enhances the tribological properties and plays a prominent role in the lubrication of cemented carbides.

Zhen et al. [98] investigated the tribological characteristics of a Ni-based solid lubricating composite containing Ag, CaF_2,_ and graphite and studied the effect of temperature on these composites from 25 °C to 800 °C in vacuum conditions. The authors revealed that the composite exhibited enhanced tribological properties over the broad temperature regime. They reported that 2 wt. % of graphite content is optimum for enhanced tribological properties in the temperature range of 25–400 °C, due to the formation of high-strength carbides. Below 600 °C, the diffusion of Ag into the worn surface helps to reduce friction and wear. Kong et al. [90] studied the HT tribological characteristics of ZrO_2_ matrix-based self-lubricating composite containing MoS_2_ and CaF_2_ as solid lubricant additives. In this experiment, they considered 10 wt. % MoS_2_ and 0–30 wt. % CaF_2_. The tribological testing was performed from RT to 1000 °C. The authors revealed that the addition of 10 wt. % MoS_2_ and CaF_2_ to the ZrO_2_ matrix manifested low friction and low wear characteristics over the broad temperature range. The reported CoF and wear rate at 1000 °C for ZrO_2_ (Y_2_O_3_)–10 MoS_2_–10 CaF_2_ was 0.27 and 1.54 × 10^−5^ mm^3^/Nm. The superior lubricating effect from RT to 400 °C was provided by MoS_2_, whereas the combined effect of CaMoO_4_ and CaF_2_ provides remarkable tribological properties from 800 °C to 1000 °C. Cui et al. [99] studied the HT self-lubricating characteristics of the CoCrW matrix with LaF_3_ and Ag as solid lubricant additives. The test parameters include a load of 10 N, a sliding speed of 0.20 m/s, Si_3_N_4_ as the counterpart, and a temperature from RT to 1000 °C using a ball-on-disk HT tribometer. The authors reported enhanced tribological properties at HT because of the combined effect of metal oxides, chromates, and LaF_3_. The reported wear mechanism at HT was abrasive and oxidative wear.

In summary, fluorides can provide superior tribological properties above 500 °C because of the brittle to ductile transition. However, at RT and temperatures below 500 °C, fluorides are brittle and cannot perform the intended task of self-lubrication. The most widely used fluorides for self-lubrication purposes are CaF_2_ and BaF_2_.

### 2.5. Hexagonal Boron Nitride (h–BN)

h–BN possesses a layered structure similar to TMDs, in which a strong covalent bond holds each layer, whereas the bonding between interlayers is the Van der Walls bond. These materials shear very quickly along the basal plane during external loading because of the lamella structure, and provide superior lubricity and enhanced tribological properties [40,100,101]. The crystal structure of h–BN consists of hexagonal rings with boron and nitrogen bonded at 120°. Lamella slip along the basal plane is considered as the prominent mechanism favoring superior lubricity [102,103]. Scholars have reported that h–BN could enhance the tribological properties of ceramics and metals [38,104,105,106]. Chen et al. [100] conducted tribological experiments on the h–BN matrix-based composites containing SiC with Al_2_O_3_ and Y_2_O_3_ as sintering additives on a rotating ball-on-disk HT tribometer from RT to 900 °C. The following paramesters were selected for the tribological study: sliding velocity of 0.094 m/s, a load of 10 N, a sliding radius of 5 mm, and a Si_3_N_4_ ball as the counterpart. They made three composites with 75, 70, and 50 vol. % of h–BN named 75BSAY, 70BSAY, and 50BSAY, respectively. Figure 13 represents the CoF of the three composites tested at different temperatures. Both 75BSAY and 70BSAY composites showed an increase in CoF with temperature rise, and the maximum values of reported CoF were 0.43 and 0.44, respectively, at 400 °C. The 70BSAY composite showed higher CoF than 75BSAY at RT and 900 °C. However, the 50BSAY composite showed higher CoF at RT, and with a rise in temperature, the CoF dropped down and reached 0.33 at 900 °C, which is 38% less than CoF at RT. The authors mentioned that adding more h–BN in the matrix can enhance the anti-friction behavior of the composites.

Cao et al. [61] conducted friction and wear tests on pure h–BN and h–BN sintered with CaB_2_O_4_ additive in atmospheric and water vapor environments from RT to 800 °C. They made two sets of composites: one with pure h–BN and the other containing 10 wt. % CaB_2_O_4_. The tribological testing was conducted on a ball-on-disk HT tribometer with the following test parameters: a load of 1.5 N, a speed of 0.188 m/s for 10 min, and Si_3_N_4_ as the counterpart. The authors observed almost similar CoF at RT for both pure h–BN and h–BN containing 10 wt. % CaB_2_O_4_ (0.18 and 0.19) tested in atmospheric conditions. Both composites showed increased CoF with temperature increases, and at 400 °C, both reported their highest CoF values of 0.58 and 0.51, respectively. The composite with 10 wt. % CaB_2_O_4_ showed a 14% reduction in CoF at 400 °C compared to pure h–BW. With a further increase in test temperature to 800 °C, the pure h–BN and the h–BN containing 10 wt. % CaB_2_O_4_ showed reduced CoF values of 0.38 and 0.35, respectively. The increased CoF with the rise in temperature from RT to 400 °C is attributed to the adhesion of h–BN on the counter ball surface, whereas the reduced CoF at HT is due to the formation of molten B_2_O_3_. However, the friction test conducted in the water vapor environment showed a reduced CoF at RT. The reported CoF values for pure h–BN and h–BN containing 10 wt. % CaB_2_O_4_ are 0.08 and 0.07, respectively. The reduced CoF is due to the formation of a water film at the interfaces under relative motion. An increase in test temperature to 400 °C for both composites led to increase in CoF (0.25 and 0.23). However, these values are lower compared to those of the composite tested under atmospheric conditions. The reduced CoF was due to the reaction of water vapor with h–BN to form B_2_O_3_ and the further reaction of B_2_O_3_ with water vapor to form H_3_BO_3_. The formed H_3_BO_3_ possessed a lamellar structure that can shear very easily under external force, which reduced the CoF. At 800 °C, the reported CoF values for pure h–BN and h–BN containing 10 wt. % CaB_2_O_4_ were 0.22 and 0.21, which are less than the CoF values observed when tribological testing was performed under atmospheric conditions. Zhao et al. [107] studied the friction and wear behavior of a nickel-based composite coating tested from 25 °C to 600 °C. The authors made three different powder compositions, which were Ni60, Ni60 with h–BN coating, and Ni60 with nano-Cu encapsulated with h–BN, represented as C1, C2, and C3. The friction and wear studies were conducted on a high-temperature pin-on-disk setup with a load of 30 N and speed of 50 rpm for 30 min using Al_2_O_3_ as the counter surface. The wear process during the tribological testing is represented in Figure 14. The common wear mechanisms reported for these coatings tested at different temperatures are abrasive and adhesive wear. The process (Figure 14, 1–4) represents the abrasive wear mechanism, which occurred under low-temperature tribological testing (25–200 °C). The strength of the coating was very high at this temperature. Under the action of external load from the Al_2_O_3_ counter body, hard particles detached from the coating. At this testing temperature, the detachment of the wear debris from the worn surface is difficult, and thus the wear rate was low.

Process 1,5,6 (Figure 14, sections 1, 5, 6) represents the mechanism of adhesive wear, which occurs at high testing temperatures, and at these temperatures, the coating has low strength. The Al_2_O_3_ counter body forcefully detaches the solid lubricant and matrix from the coating and leads to wear debris formation. The formed wear debris gets attached to the worn surface, and a glaze layer is formed. C1 underwent abrasive wear, and with the addition of Cu and h–BN, C2 and C3 underwent micro-plowing wear. At 400 °C, a combined lubricating effect from h–BN, Cr_2_O_3,_ and NiO was observed on C1 and C2, and the reported wear mechanism was a mixture of adhesive and abrasive wear. However, C3 showed adhesive wear because of CuO’s lubrication and the synergistic effects of other lubricants. Delamination was observed on the worn surfaces C1, C2, and C3 at 600 °C because of adhesive wear. Among all the coatings tested from 25 °C to 500 °C, C3 showed low CoF and wear properties. The authors revealed that nano-Cu encapsulated with h–BN increased the h–BN content in the coating, leading to enhanced wear resistance in the temperature range of 25–600 °C. Among all the coatings tested from 25 °C to 500 °C, C3 showed low CoF and wear properties. This was predominantly due to the higher h–BN content in the coating and the oxidation of copper to CuO.

In summary, h–BN possesses a layered structure, and these materials can shear very quickly along the basal plane during relative motion. h–BN possesses high thermal conductivity and oxidation resistance and is commonly used with metal matrix and ceramic matrix for self-lubricating applications because it can provide excellent tribological properties at HT conditions.

### 2.6. Polymers

Polymers are widely used in cryogenic environments because of their excellent tribological properties. For example, the mechanical components present in satellites are subjected to thermal cycling when the spacecraft is moving out of the Earth’s shadow in low-earth orbit, where the temperature is in the cryogenic range [108]. Therefore, scholars have investigated the use of polymers and self-lubricating polymer composites, especially for cryogenic temperature applications [109,110,111]. Wang et al. [109] performed tribological studies on polyimide (PI), polytetrafluoroethylene (PTFE), and polyetheretherketone (PEEK) at cryogenic temperatures in a vacuum. They observed high hardness in all the polymers in the cryogenic environment, which reduced the mating surface’s contact area. This reduction in contact area leads to reduced CoF at cryogenic temperatures. In addition, they reported a reduction in wear volume with the reduction in temperature, which is attributed to reduced mobility. Figure 15 represents the CoF and wear rate for three different polymers at −50 °C in a vacuum. The figure shows that the CoF and wear rates of polymers were lower for the 5 N than the 0.5 N load.

Theiler et al. [111] conducted tribological studies on a PEEK composite made of carbon fibers, MoS_2,_ and graphite between −80 °C and 20 °C in a vacuum environment. The authors revealed that MoS_2_-filled PEEK composite developed a smooth transfer film with high MoS_2_ concentration in the surface at −80 °C. They also revealed that sliding velocity significantly affects the tribological properties of PEEK-containing MoS_2_ solid lubricants.

Chang et al. [112] described the tribological properties of epoxy nanocomposite filled with short carbon fiber (SCF), graphite, PTFE, and nano TiO_2_ particles. The authors performed tribological testing using a pin-on-disk setup at different contact pressures and sliding velocities. A fixed contact pressure of 1 MPa and different sliding velocities, such as 0.5 m/s, 1 m/s, 1.5 m/s, and 2 m/s, were used for the first set of experiments. Tribological testing with these parameters on epoxy nanocomposites without nano TiO_2_ particles showed a CoF of 0.45 at 0.5 m/s that rose to 1 when tested at 2 m/s. However, the addition of 5 wt. % nano TiO_2_ led to a CoF of 0.3 at 0.5 m/s that rose to 0.4 at 2 m/s. In the second case, the authors used a fixed sliding velocity of 1 m/s and different contact pressures, such as 1 MPa, 2 MPa, 3 MPa, and 4 MPa. The authors observed a CoF of 0.35 at 1 MPa, which reduced to 0.25 at 4 MPa for the epoxy nanocomposite without nano TiO_2_. The authors revealed that the highest wear resistance was observed at an extreme pressure velocity (PV) factor of 12 MPa m/s, corresponding to 5 wt. % of nano TiO_2_ in the nanocomposite. Scholars also conducted tribological studies on PEEK and PEI composites containing different additives, such as SCF, graphite, PTFE, nano TiO_2_ particles, and ZnS. They performed friction and wear studies at RT and HT using different PV conditions [58,59]. The tribological test results are summarized in Table 4.

In summary, polymers are ideal solid lubricant additives for self-lubrication characteristics in a cryogenic environment, and offer excellent tribological properties. Furthermore, among polymers, PTFE provides lower CoF because of its specific structure.

### 2.7. Carbon

Carbon and carbon-based materials, such as graphite, graphene, diamond-like carbon (DLC), single-walled carbon nanotubes, multi-walled carbon nanotubes, multi-layered graphene, and graphene nanoplatelets (GNP), are used as solid lubricant additives in self-lubricating materials for challenging environments [113,114,115,116,117,118]. Among these carbon-based materials, graphene has gained tremendous attention due to its unique properties, such as high chemical inertness, high thermal conductivity, low shear strength, and enhanced mechanical and thermal properties [119]. Graphene is a two-dimensional material capable of providing low friction and wear properties, and is the fundamental building block of graphite [120]. The superior tribological properties possessed by graphite can be correlated to its layered structure, similar to MoS_2_, MoSe_2_, and WS_2_, which provides easy shearing and reduces the friction between contact surfaces in relative motion [120]. Scholars have demonstrated that graphene could be used in nano-scale or micro-scale systems to reduce friction and wear properties [121,122]. Berman et al. [120] revealed the unique behavior of graphene deposited on a steel surface when tested in dry and humid environments, and the authors observed low friction and wear. Kasar et al. [123] mentioned that graphene’s unique properties attract scholars to synthesize graphene-based self-lubricating nanocomposites for diverse applications in the automobile, aerospace, and chemical industries. Kasar et al. [124] revealed that single-layer, multilayer, and functionalized graphene could lead to reduced friction and wear rates when used as a solid lubricant additive in metal and polymer matrix composites. They summarized that graphene-based metal matrix and polymer matrix composites could be used for self-lubricating bearings. Zhai et al. [125] conducted tribological studies on Ni_3_Al matrix self-lubricating composites (NMSC) and Ni_3_Al matrix self-lubricating composites containing graphene nanoplatelets (NMSC-GNP), and explained the wear mechanism. The friction and wear studies were conducted using a ball-on-disk at HT with a load of 10 N and sliding speed of 0.2 m/s, from RT to 600 °C, with Si_3_N_4_ as the counterpart. Figure 16 represents the CoF and wear rate of NMSC and NMSC-GNP. The authors observed a CoF of 0.76 at RT for NMSC, which was reduced to 0.39 at 600 °C. NMSC−GNP exhibited a reduced CoF of 0.21–0.26 in the temperature range of RT to 400 °C, without any fluctuation, as shown in Figure 16a. However, the authors reported that at 600 °C, the CoF of NMSC−GNP increased to 0.36, close to NMSC (0.39).

The observed wear rate for NMSC was 126.8 × 10^−6^ mm^3^/N/m at RT, which reduced to 82.3 × 10^−6^ mm^3^/N/m at 400 °C. However, the reported wear rate for NMSCGNP was 5.3 × 10^−6^ mm^3^/N/m at RT, which reduced to 4.1 × 10^−6^ mm^3^/N/m at 400 °C, as shown in Figure 16b. Thus, the observed wear rate at 400 °C for NMSC-GNP is 20 times less than the corresponding wear rate for NMSC 400 °C. The wear mechanism of NMSC−GNP is shown in Figure 17. GNPs were originally distributed uniformly in the Ni_3_Al matrix, which is represented in Figure 17a. However, during the tribological testing, formation of an ultrafine-grained region on the surface was observed. This is due to the simultaneous effect of the grain refinement of GNPs and the form of wear debris containing brittle particles of Si_3_N_4_ and friable particles of NMSC, as illustrated in Figure 17b. In addition to that, slippage of the laminated sheets of GNP was observed on the worn surface, which eventually led to the formation of GNP protective layer in Figure 17c. This protective layer can reduce friction and wear rates.

Wan et al. [126] conducted tribological testing on a high−entropy alloy (HEA)-based composite and demonstrated the effects of the in situ formation of graphene on tribological properties. The composite (HEA-G) was prepared using the SPS method with graphite nanoplate (GP) as a reinforcement in the HEA matrix. The friction and wear studies were conducted using a ball-on-plate tribometer in a reciprocating mode for 1800 cycles, with a load ranging from 5 N to 100 N, a stroke length of 5 mm with various velocities, and GCr15 steel ball as the counter material. Figure 18 shows the schematic of the self-lubrication mechanism of the HEA-G composite and the formation of graphene. During tribological testing, the frictional heat caused the exfoliation of GP and the further formation of graphene. The formed graphene acts as a protective coating on the worn surface, leading to less friction and a lower wear rate. In addition to that, the tribo induced fine particles, graphene, and the fine oxide scale formed a non-continuous oxide on the worn surface. As a result, the authors observed a significant reduction in CoF and wear rate below 30 N. The superior tribological properties are attributed to the compound effect of the graphene and oxide layer formed on the worn surface.

Scholars have also adopted graphite as a reinforcement phase in metal, polymer, and ceramic matrices to develop self-lubricating materials. Graphite has also received attention for its self-lubricating and dry lubricating properties, which are attributed to the layered structure [127,128,129]. The layered structure promotes easy sliding due to the weak Van der Walls force between layers. It also possesses high thermal conductivity, which provides superior tribological properties at HT [130]. Scholars reported that the effectiveness of graphite is more pronounced in humid and air environments. Shirazi et al. [131] studied the tribological behavior of aluminum, silicon carbide, and graphite hybrid nanocomposites in atmospheric conditions and acidic solutions. They revealed that the addition of 2 wt. % graphite provided the lowest wear and CoF. Huai et al. [62] developed a graphite-based solid lubricant coating that significantly reduced CoF and wear rates tested in HT atmospheric conditions. The authors used unmodified graphite as a lubricant, amorphous SiO_2_ as the filler, and aluminum dihydrogen phosphate as a binder. The friction and wear studies were conducted using a ball-on-disk at HT with a load of 100 N and sliding speeds of 60 mm/s, 90 mm/s, and 120 mm/s, with Si_3_N_4_ as the counterpart. The tribological test was performed at 700 °C, 800 °C, and 900 °C. The authors revealed that they observed ultra-low CoF of 0.05 at 700 °C, 0.04 at 800 °C, and 0.07 at 900 °C. The authors reported that even after tribological studies, they had not observed wear scarring on the substrate surface because of the uniform coverage of the coating on the substrate. In addition to that, SiO_2_ and aluminum dihydrogen phosphate protected the graphite coating at HT, which is responsible for the superior lubricity.

In summary, carbon and carbon-based materials are promising materials for future self-lubricating applications in extreme conditions. These materials show superior lubricity, particularly under humid conditions. Hence these materials are ideal for humid environments.

## 3. Applications and Challenges

Self-lubricating materials are an advanced class of materials with diverse compositions, making them capable of performing the intended task with potential lubricating effects, especially in challenging conditions. Extreme condition lubrication is one of the potential issues faced by tribologists over the past two decades. Self-lubricants were initially introduced to enhance the efficiency and lifetime of bearings. During the early 1990s, these materials were extensively used with various mechanical components exclusively designed to operate under severe conditions to provide low friction and low wear [132]. Scholars incorporated soft metals into various matrices to give rise to self-lubricating characteristics in challenging conditions. The self-lubricating characteristics of soft metals can be attributed to the multiple slip planes, which prevent work hardening during relative motion. Usually, silver-based self-lubricating materials can provide superior friction and wear properties below 500 °C. The low shear stress and high diffusion coefficient of Ag at its interfaces provide excellent lubricity over a broad temperature range. Scholars have reported that introducing Ag and multiple solid lubricants can enhance tribological characteristics above 500 °C [64,133]. TiAl alloys are widely used for aerospace and automotive applications because of their excellent mechanical properties; however, they have inferior tribological properties. Scholars have reported that Ag addition to the TiAl intermetallic matrix enhances the tribological properties at elevated temperatures [22]. Soft metals are used in various mechanical components, such as mechanical seals, fasteners, rolling contact bearings, and sliding contact bearings, to provide superior lubricating properties. Among TMDs, MoS_2_ is considered a prospective solid lubricant capable of providing adequate lubrication over a broad temperature regime, making it specifically attractive in the aerospace, automobile, and forming industries [134]. MoS_2_ possesses outstanding friction and wear properties, which makes it a global solution for space applications. MoS_2_-based self-lubricating composites are widely used for high-vacuum and high-temperature applications. These self-lubricating composites can provide superior tribological properties, especially in the aerospace (vacuum and at HT), automotive, and forming industries (extreme pressure and temperature). The low friction and low wear properties offered by MoS_2_ depend on the external environment. The potential lubrication characteristics of MoS_2_ exist only in oxygen-free environments, and it loses its lubricating characteristics in humid and atmospheric conditions. The reported CoF for MoS_2_ in a dry or inert atmosphere is 0.002–0.05, escalating in a humid environment to 0.2 [57,135]. An exponential growth of the use of MoS_2_-based self-lubricating materials in electronic industries and battery applications has been observed in the last decade. Scholars have reported that MoS_2_ could be used for adaptive lubrication in M50 steel for aircraft bearing applications [80]. WS_2_ can function over a broad range of temperatures, including cryogenic (−190 °C) to 450 °C in air, and in extreme temperatures up to 800 °C [136]. Metallic oxide-based self-lubricants can provide remarkable tribological properties above 500 °C [137,138]. Researchers used a combination of metallic oxides and revealed that they could provide low friction and low wear characteristics at HT [139,140].

Fluoride-based solid lubricants, such as BaF_2_ and CaF_2_ dispersed in various matrices using powder metallurgy or as a coating in composites, are extensively employed in HT applications. Fluorides subjected to softening and with a smooth layer are formed at the interfaces when exposed to a temperature higher than 1000 °C. Some tri-fluorides also exhibit similar HT lubrication properties because of their chemical stability. Fluoride-based solid lubricants can be used with various matrices to provide potent lubrication properties in challenging environments. In addition, fluoride lubricants can be used in industrial applications that require self-lubrication, such as cutting tools [141], wire drawing dies [142], casting molds [143], sealing materials, and bearings [144]. They can also be used as anti-friction additives in greases and oils. h–BN can provide superior lubrication in dry and vacuum conditions [25]. It is predominantly used as a sealing coating in aircraft engines [145] PTFE-based piston rings are widely employed in reciprocating gas compressors because of their excellent self-lubrication capability and ability to form good sealing conditions [146]. PTFE-based thermoplastics are used for marine applications because of their excellent self-lubricating ability [147].

Jianxin et al. [148] developed a sintered ceramic cutting tool containing CaF_2_ for dry cutting and explained the self-lubrication behavior. They performed dry cutting on hardened steel and cast iron with Al_2_O_3_/TiC, and Al_2_O_3_/TiC with CaF_2_, at different cutting speeds. The reported CoF at the tool–chip interface for dry cutting with hardened steel was 0.65 at 60 mm/min, which rose linearly to approximately 0.75 at 80 mm/min. When CaF_2_ was added into the ceramic matrix, the CoF was reduced to 0.58 at 60 mm/min, and then declined linearly to 0.45 at 80 mm/min. When Al_2_O_3_/TiC was used for the dry cutting of cast iron, the reported CoF was approximately 0.62 at 60 mm/min, which rose linearly to approximately 0.7 at 80 mm/min. However, when the authors used CaF_2_, they observed a reduced CoF of approximately 0.64 at 60 mm/min, which declined linearly to approximately 0.56 at 80 mm/min. Thus, the enhanced tribological properties are attributed to the presence of CaF_2_ in the ceramic matrix, forming a self-lubrication film at the tool and chip interface. Niste et al. [149] studied the self-lubricating behavior of aluminum reinforced with WS_2_ composite used as a piston in automotive engines, and explained the mechanism of self-lubrication. They adopted engine operating test conditions. They used two different types of WS_2_: a flat sheet (2H–WS_2_) and an inorganic fullerene (IF–WS_2_). An aluminum composite based on IF–WS_2_ showed a CoF of 0.15, and an aluminum composite based on 2H–WS_2_ showed a CoF of 0.13, compared to the 0.29 for pure aluminum tested at 25 °C. When the tribological test was performed at 100 °C, the observed CoF values were 0.16 and 0.12 for aluminum composites based on IF–WS_2_ and 2H–WS_2_, respectively, for the first twenty minutes. The reduced CoF for 2H–WS_2_ and IF–WS_2_ in the aluminum-based composite is because of the exfoliation of the layered structure. The authors observed significant wear resistance at HT attributed to the chemical reaction of WS_2_ with the aluminum matrix to form a chemical tribo-film. Yanar et al. [150] studied the tribological behavior of low-steel composite materials containing h–BN, used as a brake pad material for railway applications. In this study, the authors used an organic brake pad with 1 wt. %, 1.5 wt. %, and 2 wt. % of h–BN. The authors observed a significant reduction in CoF when the disk surface temperature was more than 250 °C. The authors recommended 1.5 wt. % of h–BN for a stable CoF under extreme brake conditions. Yan et al. [143] conducted tribological studies on self-lubricating composite coatings containing CaF_2_ in the Co-based alloy used for casting mold applications. The scholars considered four different coatings, out of which two were dispersed with 10 wt. % and 20 wt. % of CaF_2_. The authors observed CoF values of 0.31 and 0.24 for the coating that did not contain CaF_2_. However, the coatings containing 10 wt. % and 20 wt. % of CaF_2_ possessed low CoF values of 0.19 and 0.22, respectively. The authors revealed that coatings containing 10 wt. % and 20 wt. % of CaF_2_ showed better wear resistance than other coatings. The enhanced tribological properties were due to the easy shearing of CaF_2_ along the basal plane during tribological testing. Researchers from the NASA Glenn research center developed different coatings, such as PS100, PS200, PS300, and PS400, for extreme condition applications. PS100 is a nickel–chromium-based plasma coating containing glass as a binder, and silver and fluorides are the solid lubricants. This coating has very low CoF over a broad range of temperatures and has low wear resistance. However, these coatings can be effectively applied to compressor/turbine shaft seal applications. PS200 is nickel–cobalt-based plasma coating, containing chromium carbide as a binder, and silver and fluorides are the solid lubricants. This coating has applications in the cylinder walls of Stirling engines [151]. The scholars showed that PS212 (a coating in the 200 series) could be used for foil gas bearing applications [144]. PS300 is a nickel–chromium-based plasma coating containing chromium oxide as a binder, and silver and fluorides are the solid lubricants. Researchers showed that PS304 (a coating in the 300 series) could provide low friction and a low wear rate at HT up to 650 °C [152]. PS304 is an 80% nickel/20% chromium matrix that contains solid lubricants Ag and CaF_2_/BaF_2_. Nickel and chromium offer HT oxidation resistance. For strength enhancement, chromium oxide particles were used. In addition, Ag and CaF_2_/BaF_2_ were used to provide lubrication properties at different temperature ranges. Wang et al. [153] mentioned the wearing and galling damage of lift rods of a steam turbine governor subjected to metal–metal interaction at 540 °C. The researchers applied a PS304 coating on the lift rods, and they observed that the coating on the rods was intact even after 8500 h of operation. Upon a closer examination of the coated lift rods, the authors observed the formation of a lubricious glaze film that contained Cr_2_O_3_, Ag, and CaF_2_/BaF_2_. This lubricious film prevented the galling damage of the lift rods. PS400 is a nickel–molybdenum aluminum matrix containing chromium oxide as a binder, while Ag and CaF_2_/BaF_2_ are the solid lubricants. This coating is excellent for HT wear applications, and it is used for hot foil gas bearing applications [154]. Radil et al. [155] performed tribological testing on PS400 from 260 °C to 927 °C. They observed a low CoF of 0.37 to 0.84 when tested below 927 °C, and reported that the coating was dimensionally unstable at 927 °C.

Self-lubrication is the predominant method for providing superior tribological properties in extreme condition applications. However, there are certain circumstances in which solid lubricants cannot perform their intended self-lubricating functions in certain conditions. Among the listed solid lubricants, soft metals showed softening at HT, and they became extruded from the interfaces during relative motion, limiting their self-lubricating properties. Scholars have reported that a thicker silver coating on the substrate could lead to excessive plastic deformation, resulting in increased friction. Silver diffusion at HT is another issue, and Torres et al. [156] revealed a reduction in silver at HT in self-lubricating claddings composed of silver and MoS_2_. Gao et al. [157] reported the diffusion of silver to the interface at a temperature higher than 300 °C, leading to an increased wear rate and the subsequent collapse of the coating. The diffusion can be mitigated by using a barrier against the silver migration or multilayer coating [158]. Oxidation experiments conducted at 600 °C for 100 h showed no diffusion of silver at the interface. It has been observed that with over 30 wt. % Ag addition in the coating or in the matrix, the significant Ag content in wear debris causes instability in the lubricating film [7,12]. Researchers have reported that MoS_2_ is highly susceptible to moist environments. The MoS_2_-based composite or coating gets oxidized in a humid or moist atmosphere, leading to inferior tribological properties [57]. The unsaturated bonds of MoS_2_ get exposed to H_2_O, leading to inferior tribological properties. Scholars have reported the oxidation of graphite to carbon monoxide and carbon dioxide at 400 °C and 500 °C. Significant degradation of carbon to carbon dioxide is observed beyond 700 °C [159]. The deterioration of graphite finally leads to increased pore volume, which acts as a catalyst for the oxidation reaction. These factors limit the use of graphite to moderate temperature applications. The deposition of anti-oxidant coatings and plasma deposition techniques can prevent the oxidation of graphite at HT, which is not viable from an economic point of view [160,161].

The potential issues associated with individual solid lubricants can be avoided by introducing multiple lubricants into the matrix. The introduction of multiple solid lubricants to metal, polymer, and ceramic matrix materials can have the intended effects beyond the individual self-lubrication limit. For example, Niu et al. [53] incorporated Ag and CaF_2_/BaF_2_ into the Ni_3_Al matrix self-lubricating composite, and they studied the tribological properties from 25 °C to 1000 °C. Their study revealed that from 25 °C to 400 °C, Ag provided superior lubrication characteristics. Above 400 °C, CaF_2_/BaF_2_ acts as the lubricant and provides low CoF and wear properties. At 800 °C, the molybdates that formed, such as, NiMoO_4_, BaMoO_4_ and CaMoO_4_, acted as potential lubricants and provided low CoF and wear rates.

## 4. Conclusions

In this review article, a comprehensive discussion of self-lubricating materials for extreme condition applications is provided. This review focuses explicitly on the state-of-the-art of self-lubricating materials for various challenging conditions, such as HT, cryogenic temperature, vacuum pressure, high load, high speed, and corrosive environments. Liquid lubricants do not perform well in these challenging environments and can lose their tribological properties. On the other hand, self-lubricating materials can adjust themselves based on the surroundings and provide superior tribological properties. The superior tribological behavior of these materials under extreme conditions makes these materials very popular among scientists and space engineers. The friction, wear, and lubrication mechanisms of a broad spectrum of solid lubricants dispersed in metal, polymer, ceramic, and intermetallics matrices, in various challenging environments, are explained in detail via tribological testing. In addition, recent advances and the application of self-lubricating materials have been explored. This review paper can provide deeper insights for selection of self-lubricating materials for extreme condition applications.

## Figures and Tables

**Figure 1 materials-14-05588-f001:**
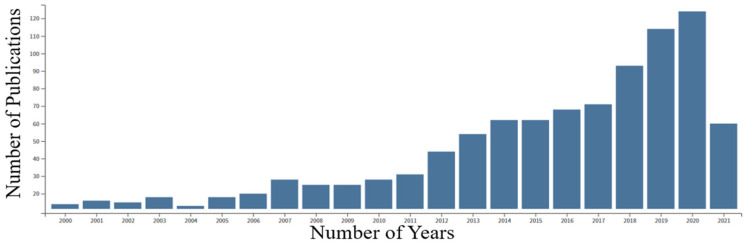
A bar graph shows journal articles published between 2000 and 2021 on “self-lubricating materials” from the Web of Science.

**Figure 2 materials-14-05588-f002:**
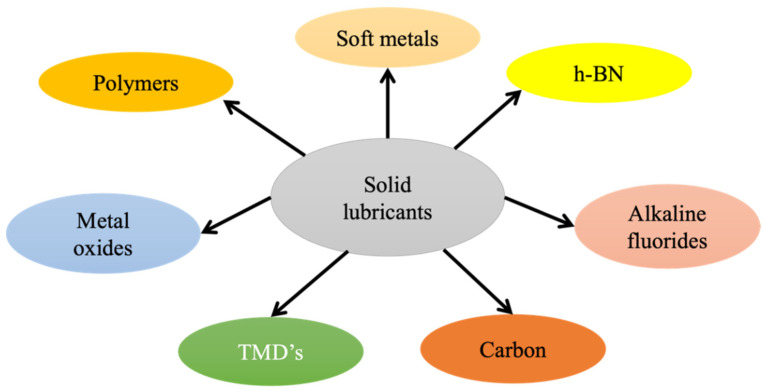
Typical solid lubricants.

**Figure 3 materials-14-05588-f003:**
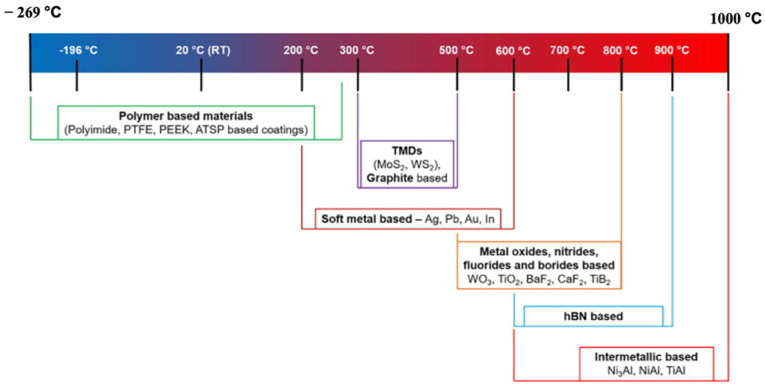
Stable working temperature for different solid lubricants. Reproduced with permission from [57]. Copyright Elsevier, 2021.

**Figure 4 materials-14-05588-f004:**
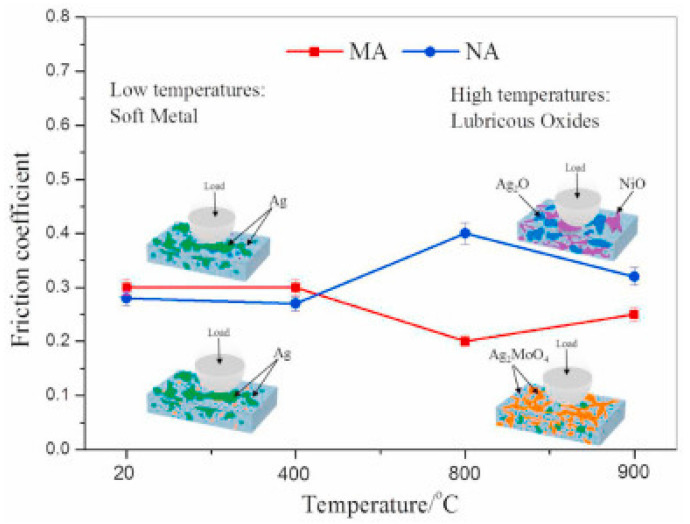
The variation in CoF with temperature and lubrication mechanism for MA and NA alloy. Reproduced with permission from [24]. Copyright Elsevier, 2021.

**Figure 5 materials-14-05588-f005:**
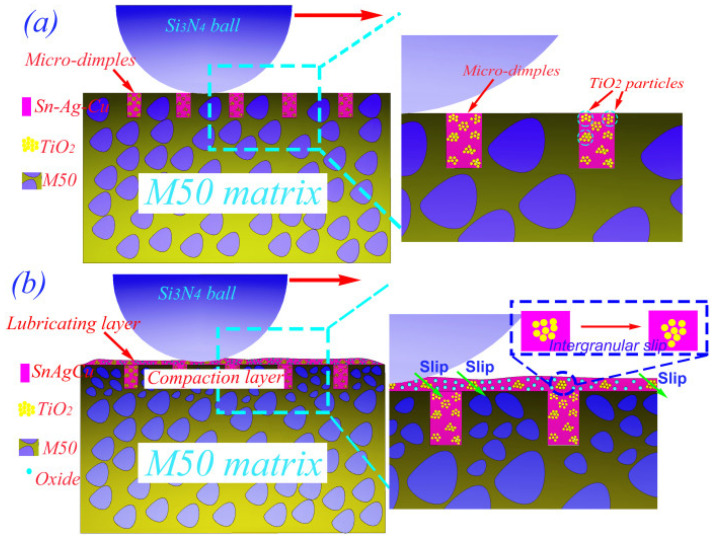
The self-lubrication mechanism of the M50 bearing dispersed with multiple lubricants (**a**) Sn, Ag, Cu/TiO_2_ removed from micro-dimples, (**b**) Slip behavior of TiO_2_ particles. Reproduced with permission from [65]. Copyright Elsevier, 2021.

**Figure 6 materials-14-05588-f006:**
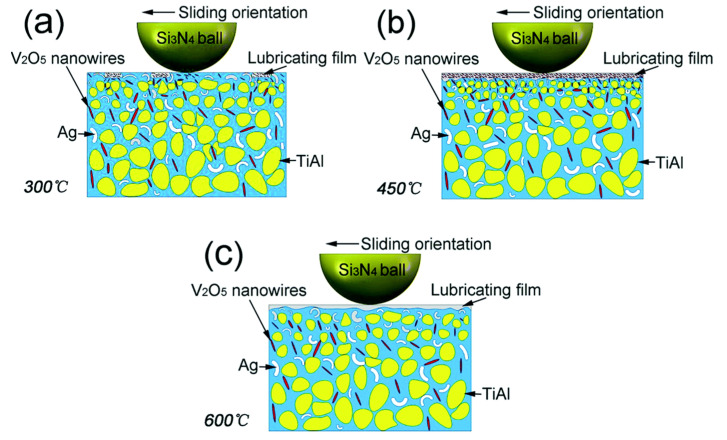
Wear mechanism of TiAl–5 Ag–1.5 V_2_O_5_ (**a**) 300 °C, (**b**) 450 °C, (**c**) 600 °C. Reproduced with permission from [66]. Copyright RCS advances, 2016.

**Figure 7 materials-14-05588-f007:**
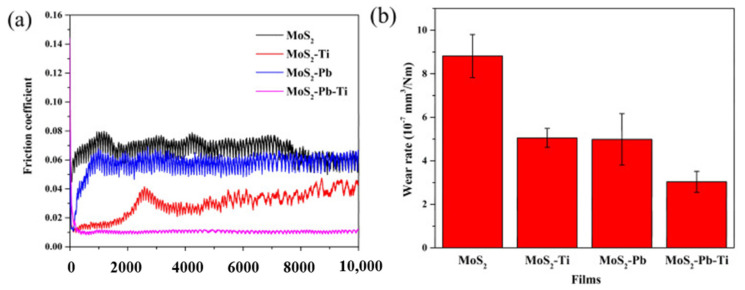
Variation in (**a**) friction coefficient and (**b**) wear rates tested in vacuum. Reproduced with permission [27]. Copyright, 2018.

**Figure 8 materials-14-05588-f008:**
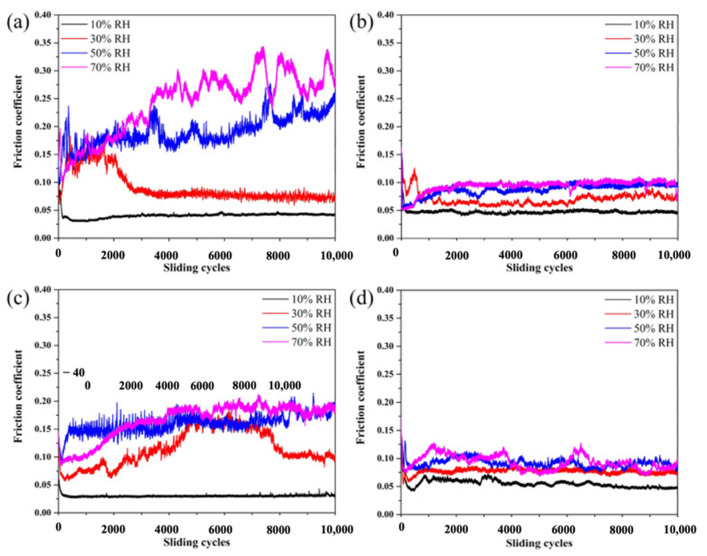
Tribological testing in the air at different RH and corresponding variation in CoF for (**a**) MoS_2_, (**b**) MoS_2_–Ti, (**c**) MoS_2_–Pb, and (**d**) MoS_2_–Pb–Ti films. Reproduced with permission [27]. Copyright, 2018.

**Figure 9 materials-14-05588-f009:**
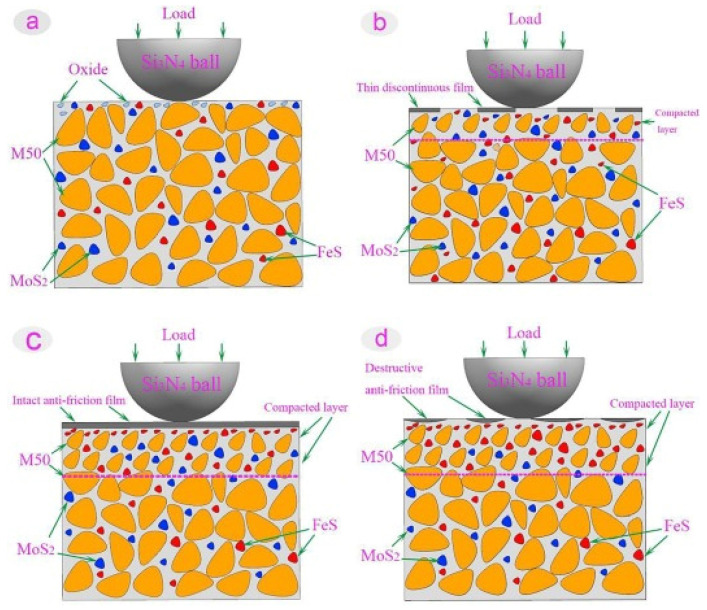
Wear mechanisms of M50 steel with 5 wt. % MoS_2_ at (**a**) 150 °C, (**b**) 250 °C, (**c**) 350 °C, and (**d**) 450 °C. Reproduced with permission from [80]. Copyright, 2017.

**Figure 10 materials-14-05588-f010:**
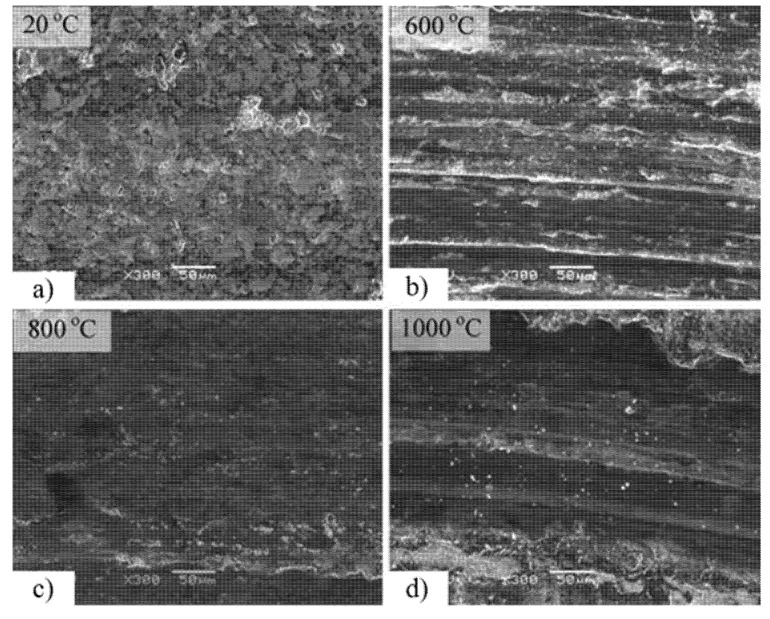
Worn surface of the composite at different temperatures (**a**) RT, (**b**) 600 °C, (**c**) 800 °C, (**d**) 1000 °C for CuO-based composite. Reproduced with permission from [29]. Copyright Elsevier, 2012.

**Figure 11 materials-14-05588-f011:**
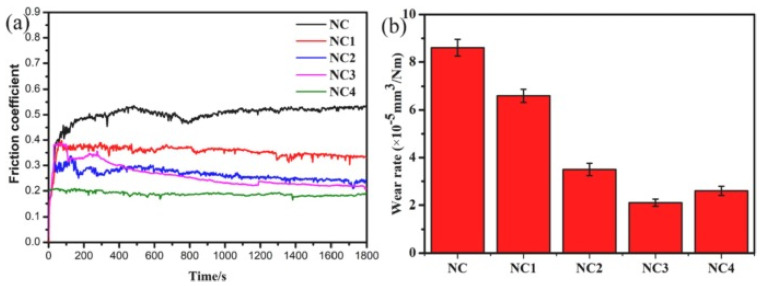
The variation of (**a**) friction coefficient, (**b**) wear rate for Ni–Cr–Mo-based composite containing TiO_2_/Bi_2_O_3_. Reproduced with permission from [87]. Copyright Elsevier, 2021.

**Figure 12 materials-14-05588-f012:**
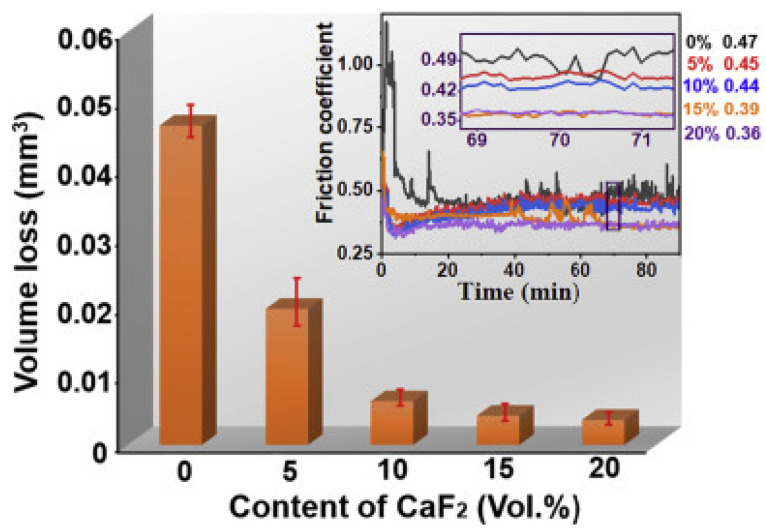
The variation in CoF and wear volume loss with different wt. % of CaF_2_ during dry sliding conditions. Reproduced with permission from [97]. Copyright Elsevier, 2021.

**Figure 13 materials-14-05588-f013:**
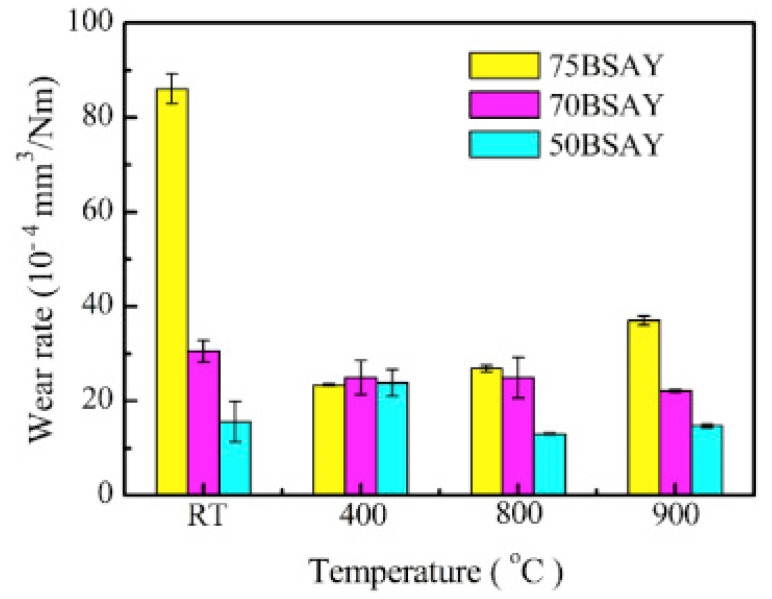
The variation in CoF for different test temperatures for the three composites. Reproduced with permission from [100]. Copyright Elsevier, 2020.

**Figure 14 materials-14-05588-f014:**
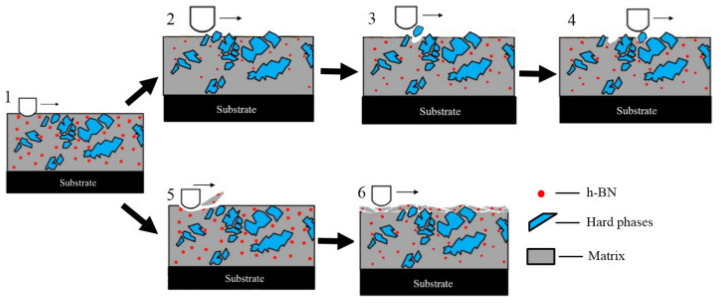
The wear process: adhesive wear observed at low temperature (1–4), abrasive wear observed at HT (1,5,6). Reproduced with permission from [107]. Copyright Elsevier, 2019.

**Figure 15 materials-14-05588-f015:**
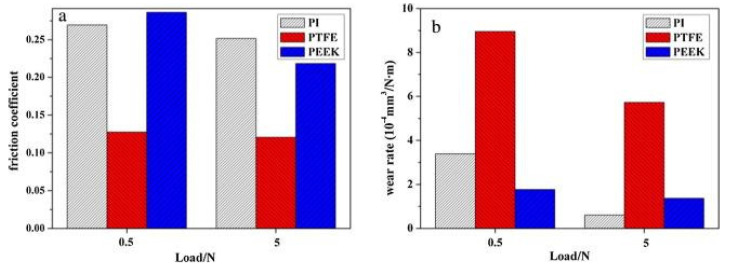
The variation in (**a**) CoF, (**b**) wear rates of three different polymer matrix-based composite tested in a vacuum at −50 °C. Reproduced with permission from [109]. Copyright Elsevier, 2016.

**Figure 16 materials-14-05588-f016:**
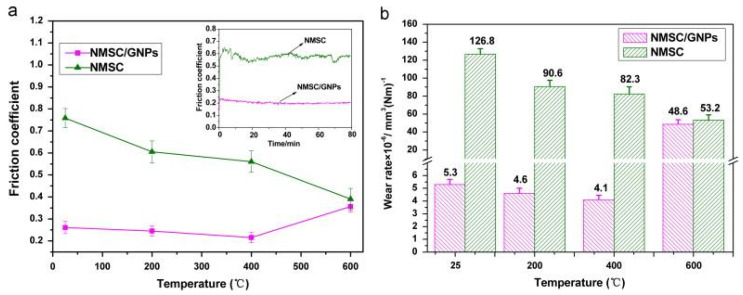
Variation in CoF of (**a**) NMSC and NMSC−GNP and wear rates of (**b**) NMSC and NMSC−GNP. Reproduced with permission from [125]. Copyright Elsevier, 2014.

**Figure 17 materials-14-05588-f017:**

The schematic of the wear mechanism of NMSC−GNP. Reproduced with permission from [125]. Copyright Elsevier, 2014.

**Figure 18 materials-14-05588-f018:**
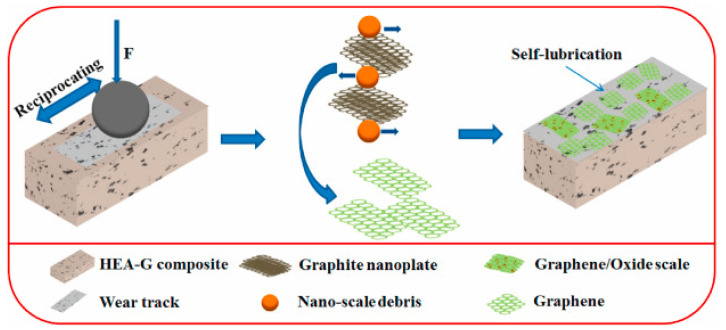
Self-lubrication mechanism of HEA-G composite. Reproduced with permission from [126]. Copyright Elsevier, 2021.

**Table 1 materials-14-05588-t001:** The applied range of extreme conditions.

Conditions	Lower Limit	Upper Limit	References
Temperature	−269 °C	1000 °C	[57]
Pressure	0.1 MPa	12 MPa	[58,59]
Humidity	10%	70%	[27]
Velocity	0.1 m/s	3 m/s	[59,60]
Load	1 N	100 N	[61,62]

**Table 2 materials-14-05588-t002:** Solid lubricants in the intermetallic matrix and their extreme condition behaviors.

Matrix/Alloy	Lubricants	Processing Route	Test Condition/Counter Body	Observation
Ni_3_Al [24]	10 wt. % and 20 wt. % Ag	High-energy ball milling (8 h) followed by vacuum hot press sintering at 900 °C (15 min, 35 MPa)	Ball-on-disk at HT; load 10 N; speed 360 r/min/Si_3_N_4_	Ag provides lubricity between RT and 400 °C Ag_2_MoO_4_ and NiO provides lubricity between 800 °C and 900 °C Silver film and oxide layers impart a synergistic effect that provides continuous lubricity over a broad range of temperatures.
Ni_3_Al [53]	Ag–Mo, and BaF_2_/CaF_2_	Vacuum hot press sintering at 1100 °C (30 min, 30 MPa)	Ball-on-disk at HT; load 20 N; sliding speed 0.20 m/s/Si_3_N_4_	Low CoF over 25 °C to 1000 °C Ag provides a lubrication effect below 400 °C BaF_2_/CaF_2_ provided the lubrication effect from 400 °C to 800 °C Molybdates provide superior lubrication above 800 °C
Ni_3_Al [54]	WS_2_, Ag, and h–BN	High-energy ball milling followed by sintering at 1150 °C (6 min, 30 MPa) in pure Ar	Ball-on-disk at HT; Load 10 N; sliding velocity 0.234 m/s/Si_3_N_4_	At 600 °C, the lowest friction and wear were reported Ag and WS_2_ provided self-lubrication at low temperature, whereas h–BN at HT
Ni_3_Al [55]	BaF_2_–CaF –Ag–Cr	Hot press sintering at 900 °C (15 min, 35 MPa)	Ball-on-disk at HT; Load 10 N; sliding velocity 0.188 m/s/Si_3_N_4_	Low CoF and wear rate observed for broad temperature range The synergistic effect of Ag, chromates, and fluorides observed at HT
Ni_3_Al [67]	BaF_2_–CaF_2_–Ag–Cr (Cr varied from 10 wt. % to 25 wt. %)	High-energy ball milling followed by hot press sintering at 900 °C (15 min, 35 MPa)	Ball-on-disk-at HT; Load 20 N; sliding velocity 0.19 m/s/Si_3_N_4_	Lowest CoF and wear rate observed for Ni_3_Al matrix with 20 wt. % Cr over broad temperature range At 800 °C, the lowest CoF and wear rate was observed 20 wt. % Cr addition is recommended for excellent lubricity
Ni_3_Al [68]	Ag, BaCrO_4_, BaMoO_4_	Ball milling followed by hot press sintering at 1100 °C (15 min, 35 MPa) powders again heated to 1200 °C for 20 min	Ball-on-disk-at HT; load 20 N; sliding velocity 0.19 m/s/Si_3_N_4_	Ag–BaCrO_4_ combination provides continuous lubrication from RT to 800 °C For the whole temperature range, the CoF observed from 0.29 to 0.38 The low CoF is because of the combined effect of Ag BaCrO_4_ and BaMoO_4_
TiAl [69]	10 wt. % Ag	Spark plasma sintering (SPS)	Ball-on-disk-at HT; load 12 N; speed 0.8 m/s/Si_3_N_4_	The primary wear mechanism was plowing from 0–30 min At 450 °C, low CoF and wear rates were observed
TiAl [22]	0 wt. %, 5 wt. %, 10 wt. %, 15 wt. % Ag	High-energy ball milling followed by sintering at 1100 °C (10 min, 30 MPa) in pure Ar	Ball-on-disk-at HT; load 10 N; speed 0.23 m/s/Si_3_N_4_	The presence of silver reduced CoF and wear RT to 800 °C showed excellent tribological properties due to Ag (moderate temperature), Ti_2_AlC, and oxides at HT 15 wt. % Ag showed improved self-lubricating behavior
TiAl [66]	5 wt. % Ag and 0.5 wt. %, 1.5 wt. %, and 2.5 wt. % V_2_O_5_ nanowires	Vibration milling followed by SPS at 1150 °C (6 min, 30 MPa) in Ar	Ball-on-disk at HT; load 20 N; sliding velocity 0.35 m/s/Al_2_O_3_	Improved tribological results were observed because of the development of the continuous film In the lubrication film, Ag provides superior lubrication while V_2_O_5_ nanowires provide shear strength The combined effect of Ag and V_2_O_5_ played a significant role at 450 °C, and the observed CoF is 0.19
TiAl [70]	Ag and Ti_3_SiC_2_	High-energy ball milling followed by SPS at 1100 °C (10 min, 35 MPa) in pure Ar	Ball on-disk at HT; Load 10 N; sliding velocity 0.23 m/s/Si_3_N_4_	Silver provided a special lubricating effect between RT and 400 *°C* Ti_3_SiC_2_ oxidation reaction forms tribo layer on the worn surface at 600–800 °C, which provides superior lubrication
TiAl [71]	12 wt. % Ag and TiB_2_ varied between 0 wt. % to 15 wt. %	High-energy ball-milling followed by SPS at 1050 °C (10 min, 35 MPa) in pure Ar	Pin-on-disk-at HT; load 12 N; sliding velocity 0.3 m/s/Si_3_N_4_	The synergistic effect of Ag and TiB_2_ provided enhanced lubrication properties at elevated temperatures TiB_2_ played the dominant role of lubrication above 600 °C compared to Ag
NiAl [72]	MoS_2,_ WS_2,_ Ti_3_SiC_2,_ and PbO	High-energy ball-milling followed by SPS at 1100 °C (5 min, 35 MPa) in pure Ar	Pin-on-disk at HT; Load 10 N; sliding velocity 0.3 m/s/Si_3_N_4_	MoS_2_ played a dominant role in self-lubrication at low and intermediate temperatures Ti_3_SiC_2_ provided superior self-lubrication at high temperature The combined effect of Ti_3_SiC_2_ and MoS_2_ showed

**Table 3 materials-14-05588-t003:** Ceramic matrix-based self-lubricating composites and their HT behavior.

Matrix/Alloy	Lubricants	Processing Route	Test Condition/Counter Body	Observation
Al_2_O_3_ [88]	Ag and CaF_2_	Powder metallurgy	Pin-on-disk; load 10 N; sliding velocity 0.168 m/s/Al_2_O_3_	Between 300 °C and 650 °C, Al_2_O_3_–50% CaF_2_ can provide moderate friction and wear rate Al_2_O_3_–20%Ag20%CaF_2_ showed a synergistic effect, which provided reduced wear and friction at HT
Al_2_O_3_ [89]	MoS_2_–BaSO_4_ doped with BaMoO_4_	Ball milling followed by SPS (1150 °C, 25 MPa, 5 min)	Standard friction and wear tester; linear stroke 1 mm; load 70 N/Al_2_O_3_	Below 200 °C, MoS_2_ lubricating film provided lower CoF BaSO_4_ and BaMoO_4_ provided low CoF and wear rates at medium and HT The synergistic effect of BaSO_4_ and BaMoO_4_, and MoO_3_ provided a low CoF at 800 °C
SiC [60]	Mo and CaF_2_	Planetary ball milling followed by hot press sintering at (1300 °C, 35 MPa, 20 min)	Ball-on-disk at HT; load 5 N; speed 0.10 m/s/SiC	Enhanced tribological properties at 1000 °C because of the presence of CaMoO_4_ on the worn surface The composite 50 SiC-20 Mo-30 CaF_2_ at 1000 °C showed the lowest CoF and wear rate
ZrO_2_ [90]	ZrO_2,_ MoS_2_ and CaF_2_	High-energy ball milling followed by hot press sintering (1200 °C, 42 MPa, 30 min)	Ball-on-disk at HT; load 10 N; speed 0.2 m/s/SiC	Tribological test performed from 200 °C to 1000 °C Up to 400 °C, MoS_2_ showed superior lubricity CaMoO_4_ with CaF_2_ formed on the wear surface provides lubricity between 800 and 1000 °C
ZrO_2_ [91]	ZrO (Y_2_O_3_)–BaCrO_4_	Ball milling followed by SPS (1500 °C, 40 MPa, 5–10 min)	Standard friction and wear tester frequency 10 Hz; linear stroke 1 mm; load 10–30 N/Al_2_O_3_	BaCrO_4_ provided superior lubricity at intermediate and HT BaCrO_4_ fine layer observed on the worn surface at HT which reduces CoF and wear rate Brittle fracture is the wear mechanism at RT

**Table 4 materials-14-05588-t004:** Tribological properties of polymer composites under different PV conditions.

Composition	PV Factors	Temperature (°C )	CoF	Specific Wear (10^−6^ mm^3^/Nm)
PEEK [58]	1 MPa, 1 m/s	RT	0.51	10.54
SCF + graphite/PEEK [58]	1 MPa; 1 m/s	RT	0.44	0.46
	2 MPa; 1 m/s		0.35	1.04
	4 MPa; 1 m/s		0.24	0.81
	2 MPa; 2 m/s		0.28	1.12
	1 MPa; 1 m/s	RT	0.41	0.53
	2 MPa; 1 m/s		0.28	0.78
	4 MPa; 1 m/s		0.23	5.76
	2 MPa; 2 m/s		0.24	3.73
PEI [59]	1 MPa, 1 m/s	RT	0.61	598.67
SCF + graphite/PEI [59]	1 MPa; 1 m/s	RT	0.56	0.77
	4 MPa; 1 m/s		0.35	1.14
	8 MPa; 1 m/s		0.22	0.73
	12 MPa; 1m/s		0.25	1.34
	4 MPa; 2 m/s		0.30	2.17
	4 MPa; 3 m/s		0.35	39.14
Nano-TiO_2_ + SCF + graphite/PEI [59]	1 MPa; 1 m/s	RT	0.36	0.30
	4 MPa; 1 m/s		0.27	2.99
	8 MPa; 1 m/s		0.15	1.62
	12 MPa; 1m/s		0.09	1.28
	4 MPa; 2 m/s		0.16	1.12
	4 MPa; 3 m/s		0.14	0.68
SCF + graphite/PEEK [58]	1 MPa; 1 m/s	70	0.14	2.19
	1 MPa; 1 m/s	150	0.46	11.01
	4 MPa; 1 m/s	150	0.44	4.66
	1 MPa; 2 m/s	150	0.54	10.03
ZnS + TiO_2_+ SCF + graphite/PEEK [58]	1 MPa; 1 m/s	70	0.08	1.04
	1 MPa; 1 m/s	150	0.15	5.38
	4 MPa; 1 m/s	150	0.12	3.5
	1 MPa; 2 m/s	150	0.14	6.38
SCF + graphite/PEI [58]	1 MPa; 1 m/s	70	0.27	4.35
	1 MPa; 1 m/s	120	0.24	1.92
	4 MPa; 1 m/s	120	0.29	2.94
	1 MPa; 2 m/s	120	0.29	15.28
TiO_2_ + SCF + graphite/PEI [58]	1 MPa; 1 m/s	70	0.09	1.74
	1 MPa; 1 m/s	120	0.08	1.08
	4 MPa; 1 m/s	120	0.1	1.56
	1 MPa; 2 m/s	120	0.26	29.27

## Data Availability

Data sharing is not applicable to this article.

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
