# Peer review of "Self-Lubricating Materials for Extreme Condition Applications"

_materials, 2021, doi:10.3390/ma14195588_

Round 1
Reviewer 1 Report
This paper reviewed the solid lubricating materials used in harsh environments and conducts a mechanism analysis in conjunction with their tribological behavior. Six types of solid lubricating materials including soft metals, transition metal dichalcogenides (TMD), metal oxides, fluorides, hexagonal boron nitride, polymers and their reinforcement in the matrix are introduced. A comprehensive review on solid lubricants dispersed in various matrices for lubrication applications, However, definition of self-adaptive materials is not clear. Also, differentiation between traditional lubricative materials and self-adaptive materials is not clear when the author organized the review article.
Other comments are listed below:
- The authors stated “conventional and liquid lubricants” in different parts, however the idea of traditional lubricants was not clear. Were the six types of the lubricants mentioned in the paper not traditional lubricants?
- There were some logic issues in current version. For example, in the abstract, the author mentioned “Self-adaptive functional materials are the most widely perceived technique for lubrication in extreme conditions in recent years”, how to understand …materials are technique…? Line 490 and 501, the descriptions were not in consist with the scheme. Line 546 and 550, “Self-adaptive lubricants were initially introduced to enhance the efficiency and lifetime of the bearings. However, during the early 1990s, these self-lubricating materials were extensively used with various mechanical components exclusively designed to operate in severe conditions to provide superior anti-friction and anti-wear properties”, no idea what is the difference between “Self-adaptive lubricants: and “self-lubricating materials”. Also, there are other inconsistence on terminology, such as “self-lubricating composite”.
- In the chapter "Applications and Challenges", not enough discussion was provided on the challenges of solid lubricating materials. The content in this chapter is still a summary of existing work and lacks prospects for the future.
- Base only on Figure 8, audience hard to understand the conclusion “The improved tribological properties underlie that Ti can significantly influence CoF than Pb when used with MoS2 and the MoS2-Pb-Ti films are an ideal candidate for humid or moist environments”, because the figure 8b and 8d did not show big difference. However, the wear rate of figure 8b was huge if one track back to the original article.
- In Figure 11, the authors mentioned “The increased wt.% of metal oxide in the matrix for these specimens reduced the CoF and wear rate, as shown in Figures 11 (a) and 11 (b)”, but it’s not clear what was the specific value of metal oxide concentration.
- There are many formatting issues, such as the inconsistency of font, color, and size between labels in different pictures. Also, for “…which is represented in Figures 10 (a) and 10 (b)…”, not letter was labeled on the figure.
In summary, this paper reviewed the solid lubricating materials used in extreme environments, and describes lubrication mechanisms according to their types. However, issues exist on the definition and concept of some idea, and lubrication mechanisms of adaptive materials for various challenging conditions were not explored in detail. Based on the above comments, I suggest a major revision of this manuscript for publication.
Author Response
Please find the rebuttal for Reviewer 1

Reviewer 2 Report
The submitted paper gives a review of the self-adaptive lubrication materials used for extreme conditions. In the reviewer´s opinion, this paper is well organized and well written. The authors give a comprehensive discussion of six types of self-adaptive materials with diverse compositions and the corresponding tribological properties. Moreover, the underlying mechanisms related to the results were also clearly described. However, the tribological properties of the formulated lubricants are evaluated mainly based on the lab tests (fundamental study). The authors are encouraged to supply further data from bench tests or practical applications. Some minor mistakes see lines 132, 133, 170, and 177.
Author Response
Please find the rebuttal for Reviewer 2

Reviewer 3 Report
- This paper mainly reviewed the tribological properties of self-lubrication materials filled by solid lubricants. It wasn’t totally different from the concept of adaption. I suggest the title can be changed “Self-lubrication materials for extreme condition applications”. But problem was recognized the second part from the type of solid lubricants, not by conditions, such as high temperature, load and speed.
- Besides, there was no literatures about pv (pressure and velocity) limit applications. You can refer the reports by Samyn Peter, Friedrich and Lanzhou institute of chemical physics, where many solid lubrication research about polymer, metal and ceramic matrix self-lubrication materials were studied for different environments and conditions.
- The applied range was also unclear, for example temperature range, the maximum load, speed, the lowest temperature resistance and high vacuum.
- The lowest wear rate was also an important parameter which determined their service life in actual application.
- Graphite and other carbon materials had excellent lubrication, why it wasn’t concluded in Figure 2? Also no SiO2. I suggest you can reclassify them.
Author Response
Please find the rebuttal for Reviewer 3

Reviewer 4 Report
The Authors presented extensive review material based on literature data (63 items). The source material is relatively modern, and the review can be considered the original study. The similarity index of the submitted material, excluding citations and bibliography, is 15% and includes mainly definitions and proper names of substances, which is fully acceptable. Therefore, there are a number of publications in the field of the submitted material. Extensive thematic monographs are also available. The Authors should therefore prove the validity of developing a review article on the problems of lubrication under "extreme" operating conditions of the friction junction. Therefore, the Authors are expected to post a few sentences after each of the subsections, in which the Authors, based on their own experience and research work, will present their own view in relation to the performance properties of each of the discussed groups of solid lubricants. Regardless of this, the Authors should propose a definition of "self-adapting greases" and present the limit values ​​of parameters defined as extreme, along with examples of friction junctions operating in extreme conditions. For formal reasons, I do not see the validity of presenting Fig. 1. This feature adds nothing to the problem at hand. The description of Fig. 14 should be supplemented with information that the presented processes relate to the use of h-BN. To sum up, in the light of the availability of publications and monographs on the problems of creating boundary layers at high loads and temperatures, I consider it necessary to supplement the material with the authors' own views on each of the groups of lubricants in question. Taking into account the specialized subject of the article, I believe that after completing the article, this article more closely corresponds to the scope of the scientific journal "Lubricants" and should be sent for publication there. I leave the decision in this matter to the Editors.
Author Response
Please find the rebuttal for Reviewer 4

Round 2
Reviewer 1 Report
No additional comment
Reviewer 3 Report
No